# Relationship between Vitreous IL-6 Levels, Gender Differences and C-Reactive Protein (CRP) in a Blood Sample of Posterior Uveitis

**DOI:** 10.3390/jcm12051720

**Published:** 2023-02-21

**Authors:** Atsushi Sakai, Mizuki Tagami, Atsuko Katsuyama-Yoshikawa, Norihiko Misawa, Yusuke Haruna, Atsushi Azumi, Shigeru Honda

**Affiliations:** 1Department of Ophthalmology and Visual Sciences, Graduate School of Medicine, Osaka Metropolitan University, Osaka 545-8585, Japan; 2Department of Ophthalmology, Kobe Kaisei Hospital, Kobe 657-0068, Japan

**Keywords:** IL-6, uveitis, diagnostic vitrectomy, case series, gender, C-reactive protein (CRP)

## Abstract

This study retrospectively determined the relationship between vitreous IL-6 levels and clinical and laboratory data collected from uveitis patients. We examined an unknown cause of posterior uveitis, collecting vitreous fluid to investigate vitreous IL-6 levels. The samples were analyzed in consideration of clinical and laboratory factors, such as the male/female ratio. The present study included 82 eyes from 77 patients with a mean age of 66.20 ± 15.41 years. The IL-6 concentrations of the vitreous specimens were 6255.0 ± 14,108.3 pg/mL in males and 277.6 ± 746.3 pg/mL in females, which was found to be a statistically significant difference (*p* = 0.048) (n = 82). There was also a statistically significant correlation between vitreous IL-6 concentrations, serum C-reactive protein (CRP) value and white blood cell counts (WBCs) (n = 82). In multivariate analysis, vitreous IL-6 levels were significantly correlated with gender and CRP in all cases (*p* = 0.048 and *p* < 0.01, respectively) and were also significantly correlated with CRP in non-infectious uveitis (*p* < 0.01). In infectious uveitis, there were no significant differences between IL-6 level and several variables. Vitreous IL-6 concentrations were higher in males than in females in all cases. In non-infectious uveitis, vitreous IL-6 levels were correlated with serum CRP. These results might suggest that intraocular IL-6 levels depend on gender differences in posterior uveitis, and intraocular IL-6 levels in non-infectious uveitis may reflect systemic inflammations, including increased serum CRP.

## 1. Introduction

There are several reasons for the occurrence of vitreous opacity, including genetic predisposition, inflammatory infectious or non-infectious conditions and the presence of degenerative and traumatic conditions, among others [1]. Uveitis is inflammation of the uvea, which consists of the iris, ciliary body and choroid. It is one of the causes of vitreous opacity, which can lead to blindness or visual impairment [2]. According to a report by Hsu et al., up to 35% of permanent visual loss cases are caused by uveitis [3]. In Japan, although there are three major causes of uveitis—ocular sarcoidosis, Vogt–Koyanagi–Harada disease and Behçet’s disease—the most frequent cause is unclassified intraocular inflammation. Unclassified intraocular inflammation (which is also referred to as idiopathic uveitis) might be associated with rheumatologic or autoimmune disease, with an estimated rate ranging from 30% to 60% of cases [4,5,6]. 

Interleukin-6 (IL-6) has been identified by Kishimoto et al. as a B-cell-stimulating factor that causes the introduction of B cells into antibody-producing cells [7]. Recently, several functions of IL-6 have been identified, with one of its major functions reported to be the development and maintenance of inflammation [8]. IL-6 enhances the expression of adhesion molecules such as VCAM-1 and ICAM-1 in endothelial cells at the inflammatory site, which then induces the production of chemokines (CXCL8/IL-8, CCL2/MCP-1 and CCL8/MCP-3) from various cells [9,10,11]. IL-6 also activates the JAK/STAT pathway, which is the reason why inflammation is maintained in several inflammatory diseases [8]. Furthermore, an IL-6 inhibitor has been recently developed and used as a treatment for rheumatoid arthritis (RA) and Castleman’s disease, with results indicating that the adaptation and use of an IL-6 inhibitor were quite effective [12].

In general, with regard to inflammation, white blood cell count and CRP levels are elevated in systemic inflammatory diseases. One study showed a significant association between serum CRP levels and a single-nucleotide polymorphism in the promoter region of interleukin-6 in Japan [13].

Other studies have reported finding an association between IL-6 and ocular disease. Murray et al. reported that aqueous humor levels of IL-6 were significantly increased in Fuchs heterochromic iridocyclitis and toxoplasma uveitis [14]. Sato et al. evaluated patients with endophthalmitis and reported on the vitreous levels for some of the cytokines, including IL-6 [15]. The vitreous level of IL-6 has also been shown to be an effective indicator in diagnosing intraocular lymphoma, which often masquerades as uveitis, thus making it difficult to diagnose [16]. When the ratio of vitreous IL-10/IL-6 is greater than 1 and heavy-chain immunoglobulin and T-cell receptor gene rearrangements are detected, this is indicative of a mean diagnosis of B- and T-cell lymphoma, respectively, with a sensitivity and specificity greater than 95% [17]. 

An additional study on uveitis patients reported some differences (age, presence of systemic disease, number of flare-ups) observed between the high-serum IL-6 group (≥5 pg/mL) and the normal-serum IL-6 group (<5 pg/mL) [6]. However, there have yet to be any reports on the relevance of vitreous IL-6 concentration in conjunction with the data collected from various patients. Thus, in the present study, we performed diagnostic vitrectomy for posterior uveitis of an unknown cause then investigated the relevance of vitreous IL-6 levels in conjunction with the manifestations and characteristics of the patients.

## 2. Materials and Methods

This retrospective observational study evaluated 82 eyes from 77 patients for unknown posterior uveitis. All subjects underwent 27-gauge diagnostic vitrectomy surgery and measurements of IL-6 concentrations at the Osaka City University Hospital between July 2018 and September 2022, and at Kobe Kaisei Hospital from October 2009 to February 2020. All patient data were retrospectively collected. After approval of the study by the Institutional Review Board at Osaka City University, Japan (IRB-4237), we obtained informed consent from all the patients enrolled in the study. 

Study inclusion criteria included [1] the existence of inflammation of the posterior vitreous, [2] unknown cause of posterior uveitis (when there was no history of Bechet disease or sarcoidosis; if ophthalmologic findings (including optical coherence tomography—OCT) and blood sampling (including HLA) did not meet diagnostic criteria, we defined this as unknown uveitis) and [3] performance of the surgery by experienced vitrectomy surgeons (surgeons who had performed more than 1000 surgeries for similar cases). Patients who did not agree to provide written informed consent were excluded from the study. Study exclusion criteria included [1] the presence of systemically active infections or diabetic retinopathy, [2] use of systemic immunosuppressive treatment such as steroids or immunosuppressive drugs for other diseases and [3] the use of immunotherapy (PD-1 inhibitors and other similar treatments) for any cancer prior to the diagnostic vitrectomy.

Vitreous samples of vitreous fluid were collected in sterile tubes and rapidly frozen at –80 °C. These samples were obtained at the time of vitreoretinal surgery.

IL-6 levels were measured in vitreous samples via enzyme-linked immunosorbent assay (ELISA) using kits for human IL-6 (Proteintech Japan, Tokyo, Japan).

Almost all of the vitreous specimens (79 out of 82 cases) underwent multiplex polymerase chain reaction (PCR) analysis for the detection of several different bacteria and viruses in accordance with previously reported methods [18]. Three specimens were obtained from a patient who had been previously diagnosed with intraocular lymphoma and was considered at the time to have a recurrence of intraocular lymphoma. Thus, this specimen was not subjected to multiplex polymerase chain reaction analysis.

Patients were assumed to have infectious uveitis when bacteria or a virus was detected in the patient’s vitreous IL-6. However, when the detection was only recognized as a result of a secondary change associated with several clinical conditions or contamination, the patient was assumed to have non-infectious uveitis [19].

When intraocular lymphoma was suspected, concentrations of interleukin IL-10 and gene arrangement studies (IgH arrangement) were considered. When an IL-10/IL-6 ratio >1 was found along with a double-positive IgH arrangement, intraocular lymphoma was indicated [17,20]. 

We also collected blood samples to measure WBC and CRP and investigated their relationship with vitreous IL-6 levels.

### Statistical Analysis

All data were collected in a Microsoft Excel spreadsheet (Microsoft, Redmond, WA, USA). Independent samples were analyzed with a *t*-test, and relevant ratios, such as the male/female ratio, were determined for each group. The Pearson correlation coefficient was used to conduct correlation analysis between IL-6 and clinical and laboratory factors. Logistic regression analysis modeling considering correlations with clinical factors was performed using SPSS (version 24, IBM Corporation, Armonk, NY, USA). A *p*-value < 0.05 was considered statistically significant.

## 3. Results

The present study examined 82 eyes from 77 patients with a mean age of 66.20 ± 15.41. (range: 14–94 years, median was 69.5 years old). In the male subjects, 35 eyes from 33 patients were evaluated, with a mean age of 67.03 ± 15.24 years (range: 16–87 years, median was 72 years old), while in the female subjects, 47 eyes from 44 patients were evaluated, with a mean age of 65.57 ± 15.51 years (range: 14–94 years, median was 67 years old) (*p* = 0.48) (Table 1). The IL-6 concentrations of the vitreous specimens were 6255.0 ± 14,108.3 pg/mL in males and 277.6 ± 746.3 pg/mL in females, which was found to be a statistically significant difference (*p* = 0.005). Table 1 presents the patient characteristics. There were 15 cases (9 males, 6 females) found to have infectious uveitis, with the detected microorganism genes being cytomegalovirus (n = 3), human T-cell leukemia virus type 1 (HTLV-1: n = 3), Epstein–Barr virus (n = 2), varicella zoster virus (n = 2), bacterial 16-strip PCR (n = 2), herpes simplex virus type 1 (n = 1), human herpesvirus 6 (n = 1) and *Propionibacterium acnes* (n = 1). (raw data: Appendix A).

There were 67 cases (26 males, 41 females) of non-infectious uveitis. IL-6 concentrations were compared between males and females for the infectious and non-infectious uveitis cases. The IL-6 concentration in the infectious uveitis group was 9023.1 ± 12,792.8 pg/mL in males versus 87.5 ± 99.3 pg/mL in females (*p* = 0.26). The IL-6 concentration in the non-infectious uveitis group was 5296.8 ± 14,412.6 pg/mL in males versus 305.4 ± 794.4 pg/mL in females (*p* = 0.033).

Figure 1 shows scatter plots for vitreous IL-6 concentration versus age, WBC and CRP overall and in infectious/non-infectious uveitis. In all cases (a~c), there were significant correlations between vitreous IL-6 concentrations and WBC (*p* < 0.01) and CRP (*p* < 0.01) but not age (*r* = −0.02, *p* = 0.84). In infectious uveitis (d~f), there were no significant correlations between IL-6 level and WBC (*r* = −0.03, *p* = 0.91), CRP (*r* = 0.07, *p* = 0.80) or age (*r* = 0.15, *p* = 0.60). In non-infectious uveitis (G–I), significant correlations between vitreous IL-6 level, WBC (*r* = 0.50, *p* < 0.001) and CRP (*r* = 0.81, *p* < 0.01) were found but not age (*r* = −0.06, *p* = 0.61). 

We also performed multiple regression analysis for all cases in both infectious uveitis and non-infectious uveitis patients (Table 2). There were significant differences between gender (male dominant) and CRP in all cases (*p* = 0.048 and *p* < 0.01, respectively), and there was also a significant difference in CRP in non-infectious uveitis (*p* < 0.01). In infectious uveitis, there were no significant differences between IL-6 level and several variables.

## 4. Discussion

IL-6 is a cytokine that plays an important role in the regulation of immune and inflammatory responses via the activation of the JAK/STAT pathway. Increases in the IL-6 level are known to be related to the occurrence of many autoimmune and inflammatory diseases, such as rheumatoid arthritis (RA), juvenile idiopathic arthritis (JIA) and Castleman’s disease [8]. The IL-6 inhibitor tocilizumab has been shown to be effective in decreasing this inflammation, and thus, it has recently become widely used in these patients. 

In this study, we found significant correlations between gender and vitreous IL-6 levels in all cases. No significant changes in the parameters were found in infectious uveitis cases; however, non-infectious uveitis cases demonstrated a significant correlation between CRP and vitreous IL-6 levels. These results might suggest that intraocular IL-6 levels depend on gender, and non-infectious uveitis may lead to systemic inflammations due to intraocular IL-6 [12].

Okada et al. reported a significant association between a single-nucleotide polymorphism (rs2097677) in the promoter region of interleukin 6, a representative multifaceted inflammatory cytokine, and serum CRP levels in Japanese subjects [13]. Considering this study, there may be a relationship between vitreous IL-6 concentration and serum CRP expression levels in posterior uveitis as well.

There have been other reports in other scientific fields with regard to differences in IL-6 expression between males and females. Andreas et al. reported that male patients with septic and multiple-organ failure after severe injuries (defined as an Injury Severity Score over 25) tended to have significantly higher plasmic IL-6 levels as compared to females [21]. Another report evaluated blood IL-6 concentrations in patients with COVID-19 infections. Since it is known that males have a higher rate of severe COVID-19 and COVID-19 fatalities as compared to females, Emily et al. further examined these types of patients and reported that various inflammatory cytokines, such as IL-6 and C-reactive protein, were predominantly higher in males with COVID-19 at the time of hospitalization and during peak infection levels [22]. Based on our knowledge of the differences in the levels of IL-6 between males and females, these findings might suggest that there could be a potential relationship between sex hormones and IL-6 levels.

Although not directly related to the ophthalmological field, there are indeed reports that have examined the relationship between IL-6 levels and sex hormones. Wei et al. reported that in patients with hepatitis-B-virus-associated hepatocellular carcinoma, estrogen binds to intracellular estrogen receptors in Kupffer cells, which inhibits the STAT3 and NFκB pathways, which are required in order to activate the IL-6 promoter [23]. When these pathways were inhibited, there was a decrease in IL-6 expression in conjunction with a decrease in damage to liver cells.

In endotoxin-induced uveitis in rats, estrogen reduces the expression of E-selectin and the IL-6 level, which then causes the suppression of cellular infiltration [24]. Sandra et al. reported that the estrogen receptor alpha (ERα) is expressed in the retinal pigment epithelial cells in young women but not in men or in postmenopausal women; [25] in addition, they also reported that ERα was expressed in the ciliary body, iris and in the epithelium of the lens.

When taking these reports into consideration, we might suggest that estrogen reduces IL-6 expression in the human eye, thereby reducing intraocular inflammation. If estrogen reduces IL-6 levels, there could also potentially be differences in the vitreous IL-6 concentrations between premenopausal and postmenopausal females. In non-infectious uveitis, intraocular IL-6 is correlated with systemic inflammation; this does not contradict the real-world findings evidencing the usefulness of IL-6 inhibitors for non-infectious uveitis [26]. However, this is only a hypothesis, because the average age of the women in this study was high, and blood estrogen levels were not measured, so further investigation is needed.

There were some limitations to our present study. First, this was a retrospective study. In addition, we evaluated a limited number of patients (N = 82) for the relationship between IL-6 levels and gender differences. In order to more specifically evaluate this relationship, we will need to increase the number of subjects and examine more cases prospectively. In addition, ocular sarcoidosis often presents nonspecific findings, so some ocular sarcoidosis cases could be classified as unknown uveitis.

Another limitation of our study concerns the relationship between estrogen and IL-6. In the present study, we did not examine the blood estrogen level, and as a result, it is difficult to strictly evaluate the relevance of our present findings. Thus, when performing any future investigations, it will be important to measure not only the vitreous IL-6 levels but also the blood IL-6, androgen and estrogen levels and then compare our results.

In conclusion, the vitreous IL-6 level was significantly higher in male patients, and non-infectious uveitis cases showed a significant correlation between CRP and vitreous IL-6 levels. These results suggest that intraocular IL-6 levels depend on gender differences, and the intraocular IL-6 levels in non-infectious uveitis may reflect systemic inflammations.

Further investigations into the relationship between vitreous IL-6 levels and clinical parameters including gender and CRP for the diagnosis and prognosis of posterior uveitis are needed.

## Figures and Tables

**Figure 1 jcm-12-01720-f001:**
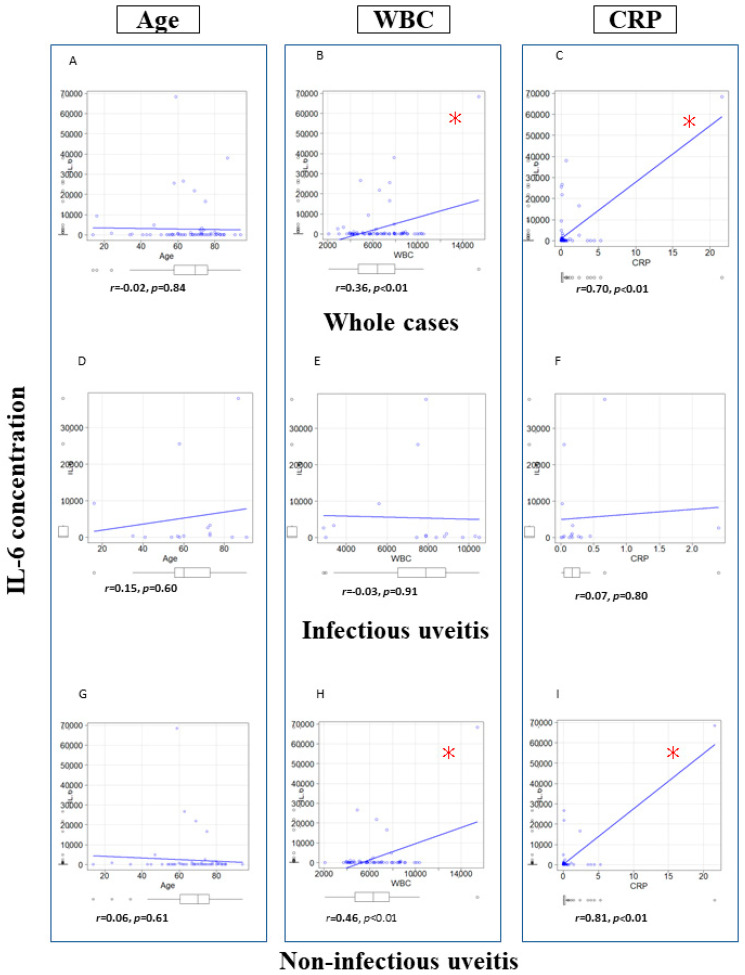
Scatter plots depicting vitreous IL-6 concentration correlations with age, WBC and CRP. (**A**): Correlation of IL-6 and age in whole cases. (**B**): Correlation of IL-6 and WBC in whole cases. (**C**): Correlation of IL-6 and CRP in whole cases. (**D**): Correlation of IL-6 and age in infectious uveitis. (**E**): Correlation of IL-6 and WBC in infectious uveitis. (**F**): Correlation of IL-6 and CRP in infectious uveitis. (**G**): Correlation of IL-6 and age in non-infectious uveitis. (**H**): Correlation of IL-6 and WBC in non-infectious uveitis. (**I**): Correlation of IL-6 and CRP in non-infectious uveitis. * *p* < 0.01. Pearson correlation coefficient. WBC: White blood cell, CRP: C-reactive protein, IL-6: Interleukin-6.

**Table 1 jcm-12-01720-t001:** Male and female patient characteristics.

	Male	Female	
Patients	33	44	
Eyes	35	47	
Age	63.0 ± 19.26	58.4 ± 16.60	*p* = 0.677
IL-6 levels	6255.0 ± 14,108.3	277.6 ± 746.3	*p* = 0.005
Unilateral	31	41 (1 case is recurrence)	
Bilateral	2	3	
Infectious uveitis	9	6	
IL-6 levels	9023.1 ± 12,792.8	87.55 ± 99.34	*p* = 0.135
Non-infectious uveitis	26	41	
IL-6 levels	5296.8 ± 14,412.6	305.4 ± 794.4	*p* = 0.03
Lab data			
WBC (/μL)	6298.6 ± 2504.0	6338.1 ± 1868.5	*p* = 0.231
CRP (mg/L)	1.30 ± 3.72	0.23 ± 0.56	*p* = 0.057

IL-6: Interleukin-6.

**Table 2 jcm-12-01720-t002:** Multivariate regression analysis: correlations between IL-6, age, gender, WBC and CRP.

A. In All Cases (N = 82).			
Variable	Regression Coefficient	95% CI	*p*-Value
Age	−24.07	−128.16–80.00	0.65
Gender(male)	−3254.23	−6473.73–−34.72	0.048 *
WBC	0.11	−0.70–0.93	0.78
CRP	2500.15	1794.47–3205.82	<0.01 *
B. In infectious uveitis cases (N = 15).			
Variable	Regression coefficient	95% CI	*p*-value
Age	130.50	−239.70–500.70	0.45
Gender	−10,253.93	−24,931.07–4425.21	0.15
WBC	0.53	−2.63–3.71	0.71
CRP	−1006.21	−14,127.39–12,114.98	0.87
C. In non-infectious uveitis cases (N = 67).			
Variable	regression coefficient	95% CI	*p*-value
Age	−48.38	−145.47–48.71	0.32
Gender	−1565.76	−4459.90–1328.38	0.28
WBC	−0.15	−0.94–0.65	0.71
CRP	2734.08	2127.56–3340.61	<0.01 *

* logistic regression analysis.

## Data Availability

All data generated or analyzed during this study are included in this published article.

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
