# Peer review of "Relationship between Vitreous IL-6 Levels, Gender Differences and C-Reactive Protein (CRP) in a Blood Sample of Posterior Uveitis"

_jcm, 2023, doi:10.3390/jcm12051720_

Round 1
Reviewer 1 Report
Dear colleagues,
congratulate you on a good idea for research, but a few changes and additions are needed for the ultimate good paper.
1. The title should be changed!
I guess the title should have read "Relationship of vitreous IL-6 Levels with gender difference and C-reactive protein (CRP) in a blood sample (vitreous sample) of posterior uveitis.
2. There is some mistake in row 100: IL-10/IL-6 ratio >10
3. In Materials and methods:
- the blood sample collection method is not described
- diabetic retinopathy should be an exclusion criterion
4. In discussion:
Although it is a good assumption that estrogens reduce Il-6 concentration in women, it should be explained that this statement is not completely convincing considering the average age of the women in the study (which could be confirmed by measuring the level of estrogen in the blood).
Author Response
Response to reviewers
Thank you your important advices.
Reviewer1
- The title should be changed!
I guess the title should have read "Relationship of vitreous IL-6 Levels with gender difference and C-reactive protein (CRP) in a blood sample (vitreous sample) of posterior uveitis.
→We changed the title as below,
“Relationship of vitreous IL-6 Levels with gender difference and C-reactive protein (CRP) in a blood sample of posterior uveitis.”
Thank you for your advice.
- There is some mistake in row 100: IL-10/IL-6 ratio >10
→We corrected to “IL-10/IL-6 ratio >1” (P3, L107), thank you for your remarks.
- In Materials and methods:
- the blood sample collection method is not described
→ “We also collected blood samples to measure WBC and CRP at the first visit to our department, and we investigated the relationship between vitreous IL-6 level and them in blood.” (P3, L110 to 111)
We added above texts, we appreciate for your advice.
- diabetic retinopathy should be an exclusion criterion
→We added diabetic retinopathy as an exclusion criterion (P2, L88 to 89), thank you for your remarks.
- In discussion:
Although it is a good assumption that estrogens reduce Il-6 concentration in women, it should be explained that this statement is not completely convincing considering the average age of the women in the study (which could be confirmed by measuring the level of estrogen in the blood).
→” However, this is just a hypothesis, because average age of the women in this study was high, and blood estrogen levels were not measured, so further investigation is needed.” (P7, L242 to 244)
We added above sentence, thank you for your important advice.
Reviewer 2 Report
General comments:
The authors have submitted a retrospective study describing the relationship between vitreous IL-6 levels in different genders and with different CRP levels. While the authors have tried to explain the rationale for measuring IL-6 levels they do not clearly provide the same for age, CRP levels and WBC. It is therefore not clear to the readers why those specific variables were chosen. In the opinion of this reviewer, lines 196 to 212 should rather be included in the introduction as it provides the rationale for studying CRP levels. The reason for studying WBC counts remains unclear.
Specific comments:
1. In line 78 the authors state that one inclusion criterion was 'unknown cause of posterior uveitis'. It is important to inform the readers how other causes of uveitis were excluded. In lines 89 to 91 they mention that multiplex PCR was performed on 79/82 patients, which is relevant, but no indication is given of how VKH, Behcet's disease and sarcoidosis were excluded and these are the most common causes of uveitis where they see patients.
2. In line 57 the authors mention an IL10/IL6 ratio >1 but in line 100 they chose an IL10/IL6 ratio >10. Can the authors please clarify the discrepancy?
3. In the results section, please provide the median age(s) as well so that the readers may work out the age distribution.
4. In line 118 the p-value is given as 0.045 but in Table 1 it is given as 0.048. Which is correct? Given the large difference in means between males and females one might have expected an even smaller p-value. Please double check all the p-values throughout.
5. In line 91, the authors cite reference 17 as previously reported methods. According to ref 17 the strip PCR tests for HSV1, HSV2, varicella-zoster virus, human T-cell lymphotropic virus 1, human herpesvirus 6, Epstein-Barr virus, cytomegalovirus, Toxoplasma gondii, and Treponema pallidum. However, in the results they report bacterial 16-strip PCR (n=2) and Propionibacterium acnes (n=1) which were not part of the multiplex strip PCR. In reference 17, they cited a reference 13 which reported a multiplex strip PCR which tested for 24 pathogens including bacterial 16S and P. acnes. This is bound to confuse the readers and needs to be clarified please.
Author Response
Reviewerï¼’
Thank you your important advices.
General comments:
The authors have submitted a retrospective study describing the relationship between vitreous IL-6 levels in different genders and with different CRP levels. While the authors have tried to explain the rationale for measuring IL-6 levels they do not clearly provide the same for age, CRP levels and WBC. It is therefore not clear to the readers why those specific variables were chosen. In the opinion of this reviewer, lines 196 to 212 should rather be included in the introduction as it provides the rationale for studying CRP levels. The reason for studying WBC counts remains unclear.
→We mentioned why we measured CRP and WBC as below at P2, L 51-54.
“In general, with regard to inflammation, blood WBC and CRP level are elevated in systemic inflammatory diseases. There is a study which showed that a significant association between serum CRP levels and a single nucleotide polymorphism in the promoter region of interleukin-6 in Japanese.(13)”
Thank you for your important advice.
Specific comments:
- In line 78 the authors state that one inclusion criterion was 'unknown cause of posterior uveitis'. It is important to inform the readers how other causes of uveitis were excluded. In lines 89 to 91 they mention that multiplex PCR was performed on 79/82 patients, which is relevant, but no indication is given of how VKH, Behcet's disease and sarcoidosis were excluded and these are the most common causes of uveitis where they see patients.
→Thank you for your advice.
Certainly, a definitive diagnosis of non-infectious uveitis can only be made by a combination of several ophthalmologic and systemic examinations.
Additionally described in Materials and methods,
Discussion may refer to ocular sarcoidosis as undiagnosed non-infectious uveitis.
(when there were no history of Bechet disease and sarcoidosis, and ophthalmologic find-ings (including Optical Coherence Tomography: OCT) and blood sampling (including HLA) were not meet for diagnostic criteria, we defined them as unknown uveitis) (L82)
In addition, this is about inclusion criteria, ocular sarcoidosis often presents nonspecific findings, so some ocular sarcoidosis cases might slip into unknown uveitis. (L248)
- In line 57 the authors mention an IL10/IL6 ratio >1 but in line 100 they chose an IL10/IL6 ratio >10. Can the authors please clarify the discrepancy?
→We corrected to “IL10/IL6 ratio >1”, (P3, L107) we appreciate to your remarks.
- In the results section, please provide the median age(s) as well so that the readers may work out the age distribution.
→ We added the median age in the results section(P3 ,L122 to 129) as below, thank you for your advice.
・The present study examined 82 eyes of 77 patients with a mean age of 66.20 ±
15.41(range: 14–94 years, median was 69.5 years old). (P3, L122)
・In the male subjects, 35 eyes of 33 patients were evaluated, with a mean age of 67.03 ± 15.24 years (range: 16-87 years, median was 72 years old), (P3, L123-124)
・in the female subjects, 47 eyes of 44 patients were evaluated, with a mean age of 65.57 ± 15.51 years (range: 14–94 years, median was 67 years old). (P3, L124-125)
- In line 118 the p-value is given as 0.045 but in Table 1 it is given as 0.048. Which is correct? Given the large difference in means between males and females one might have expected an even smaller p-value. Please double check all the p-values throughout.
→Thank you for your very important point.
When the non-pair-t test was performed, P < 0.01 as you pointed out. I was correcting table1 immediately. (Table.1 gender male vs female P=0.005)
- In line 91, the authors cite reference 17 as previously reported methods. According to ref 17 the strip PCR tests for HSV1, HSV2, varicella-zoster virus, human T-cell lymphotropic virus 1, human herpesvirus 6, Epstein-Barr virus, cytomegalovirus, Toxoplasma gondii, and Treponema pallidum.However, in the results they report bacterial 16-strip PCR (n=2) and Propionibacterium acnes (n=1) which were not part of the multiplex strip PCR. In reference 17, they cited a reference 13 which reported a multiplex strip PCR which tested for 24 pathogens including bacterial 16S and P. acnes. This is bound to confuse the readers and needs to be clarified please.
→Certainly, we confirmed that the paper we cited (reference 17, which is reference No.18 after correction) had referred to reference 13 in the paper.
As you told us, the paper we used to cited had not reported bacterial 16-strip PCR and Propionibacterium acnes. We think it is better to cite reference 13 in the paper, too. So, we replaced the paper to cite.
Thank you for your very important remarks.
Round 2
Reviewer 2 Report
The authors have addressed all my concerns in a satisfactory manner.